# Extraction and Purification of Flavonoids and Antiviral and Antioxidant Activities of *Polygonum perfoliatum* L.

**DOI:** 10.3390/molecules30010029

**Published:** 2024-12-25

**Authors:** Chan Zhao, Jian Xu, Yao Liu, Peng Xu, Jin Yi, Lin Feng, Yanyan Miao, Yongping Zhang

**Affiliations:** 1College of Pharmaceutical Sciences, Guizhou University of Traditional Chinese Medicine, Guiyang 550025, China; zc3523711813@126.com (C.Z.); twt8489@126.com (J.X.); 13668500680@163.com (Y.L.); 13195264157@163.com (P.X.); 18270564511@163.com (J.Y.); fenglin28@126.com (L.F.); 2National Engineering Technology Research Center for Miao Medicine, Guiyang 550025, China; 3Guizhou Engineering Technology Research Center for Processing and Preparation of Traditional Chinese Medicine and Ethnic Medicine, Guiyang 550025, China

**Keywords:** *Polygonum perfoliatum* L., flavonoids, extraction, purification, antiviral, antioxidant

## Abstract

The aim of the present study was to optimize the process parameters for the extraction and purification of total flavonoids from *Polygonum perfoliatum* L., in addition to analyzing their chemical composition and evaluating their activity against varicella-zoster virus (VZV) and antioxidant activity. The optimum extraction process was determined using one-way and response surface methods with the following conditions: ethanol concentration of 82.00%, temperature of 90.29 °C, solid-to-liquid ratio of 1:32.78 g/mL, extraction time of 1.5 h, and two extractions with a yield of 14.98 ± 0.11 mg/g. Purification was then carried out using D101 macroporous resin to obtain a flavonoid purity of 43.00 ± 2.55%, which was 3.38 times higher than that of the crude extract (12.74 ± 1.04%). Further purification was carried out using a 1:9 hybrid column of macroporous resin and polyamide, and the purity of flavonoids was enhanced to 59.02 ± 2.23%, which is 1.37 times higher than that of the macroporous resin purifier (43.00 ± 2.55%) and 4.63 times higher than that of the crude extract (12.74 ± 1.04%). Seventy-nine flavonoids were identified using ultra-performance liquid chromatography-tandem high-resolution mass spectrometry (*UPLC-HRMS*). In addition, the purified flavonoids showed good anti-VZV and antioxidant activities. Therefore, this study can provide technical support and theoretical basis for the further development and utilization of *Polygonum perfoliatum* L. resources.

## 1. Introduction

*Polygonum perfoliatum* L., an annual herb in the Polygonaceae family, is native to various regions, including China, Korea, Japan, Indonesia, the Philippines, and India [1]. It serves multiple purposes; it is not only used as a culinary ingredient and poultry feed but is also valued for its high medicinal potential [2]. Studies have revealed that *Polygonum perfoliatum* L. is rich in a range of bioactive compounds, including flavonoids, phenylpropanoids, anthraquinones, terpenoids, and steroids [3]. These constituents exhibit various pharmacological effects, such as antiviral, anti-inflammatory, hepatoprotective, anticancer, and antibacterial activities [1].

Flavonoids are a group of secondary metabolites that are widely distributed in nature, known for their significant biological activities. They are also one of the main active components of *Polygonum perfoliatum* L. [4,5]. To maximize the yield of total flavonoids in *Polygonum perfoliatum* L., Zhang et al. employed a water decoction method combined with an orthogonal test to select the optimal extraction process. However, the yield of flavonoids was only 7.21 mg/g [6]. Huang et al. used a microwave-assisted method to optimize the extraction of total flavonoids from *Polygonum perfoliatum* L., achieving an average yield of 5.69% [7]. While this method has contributed to the exploitation of flavonoids from *Polygonum perfoliatum* L., microwave extraction faces challenges such as limited industrial application and concerns over operational safety [8,9], restricting its large-scale utilization. In contrast, the reflux extraction method offers advantages such as high solvent utilization, relatively simple operation, and suitability for industrial production [10]. Response surface methodology (RSM) is a statistical technique used to optimize and analyze the relationship between multiple independent and dependent variables through the construction of a mathematical model. This method is particularly useful for determining optimal response values through a smaller number of experiments, making it flexible and cost-effective. RSM has been widely used in various extraction methods in traditional Chinese medicine, such as the microwave-assisted extraction method and reflux extraction method. [11,12]. For instance, Hu et al. optimized the reflux extraction process to maximize the yield of Notoginseng Radix et Rhizoma saponins using RSM [13] and, similarly, Yang et al. employed this methodology to optimize the reflux extraction of total triterpenoid saponins from Celosiae Semen.

Purification of flavonoids is essential due to their strong pharmacological activity and the low yield obtained from *Polygonum perfoliatum* L. Various techniques have been employed for flavonoid purification, including macroporous resin adsorption, polyamide adsorption, recrystallization, solvent extraction, and membrane separation methods [14]. Among these, macroporous resin adsorption stands out for its advantages of being non-polluting and offering efficient separation, making it one of the most commonly used techniques [15]. For instance, macroporous resins have been successfully used to enrich and purify flavonoids from Abelmoschus manihot flowers and Flos Populi extracts, significantly improving their purity [16,17]. Additionally, polyamides, due to their amide groups, can form hydrogen bonds with phenolic hydroxyl groups, exhibiting excellent adsorption properties that make them effective in flavonoid purification [18]. Li et al. reported that polyamide purification increased the purity of total flavonoids in Ginkgo biloba flowers to 37.1 ± 1.1% [19]. Similarly, flavonoids from taxoid-free Taxus residue extracts were purified using polyamides, raising the flavonoid content from 20.65% to 65.21% [20]. Compared to single-packed chromatographic columns, double-packed hybrid chromatographic columns offer improved purification efficiency, reducing process steps and enhancing the overall purification effect. Yu et al. demonstrated this by mixing HPD600-type and AB-8-type macroporous resins at a 3:2 ratio, achieving a purity increase in total flavonoids from *Gynura divaricata* (L.) DC. from 12.67% to 70.63% [21].

The *varicella-zoster virus* (VZV) is a pathogenic human herpesvirus. Its initial infection manifests as chickenpox, after which the virus can remain latent in the peripheral ganglia. Upon reactivation, it causes herpes zoster (shingles) [22]. Currently, the treatment for the acute phase of herpes zoster primarily relies on oral antiviral drugs, such as acyclovir, and analgesics. However, these treatments often lead to toxic side effects, drug resistance, and complications such as postherpetic neuralgia [23]. As a result, finding a safe, cost-effective, and efficient treatment has become a priority. Traditionally, *Polygonum perfoliatum* L. has been used for the treatment of herpes zoster, with its fresh juice crushed and applied directly to the affected area [1]. Zhang et al. demonstrated that flavonoids can inhibit viral replication and cell-to-cell transmission. Moreover, the antiviral activity of *Polygonum perfoliatum* L. was confirmed through in vitro and in vivo experiments targeting HSV-1 infection [24].

In this study, the response surface method was used for the first time to accurately optimize the process parameters for the reflux extraction of *Polygonum perfoliatum* L. flavonoids on the basis of one-way experiments, which improved the yield of *Polygonum perfoliatum* L. flavonoids and made them suitable for industrial production. Subsequently, we innovatively used a mixed column filled with macroporous resin and polyamide to further optimize the process parameters for purifying the total flavonoids of *Polygonum perfoliatum* L. In addition, the purified flavonoids of *Polygonum perfoliatum* L. were characterized in depth using ultra-performance liquid chromatography-tandem high-resolution mass spectrometry (*UPLC-HRMS*), which significantly improved the accuracy and efficiency of the compositional identification. Finally, we evaluated the anti-VZV and antioxidant activities of the purified flavonoids of *Polygonum perfoliatum* L., which not only provided a scientific basis for the in-depth research and application of *Polygonum perfoliatum* L., but also laid a solid foundation for its application in medicine and health products.

## 2. Results and Discussion

### 2.1. Results of One-Way Experiments on Flavonoid Extraction

#### 2.1.1. Ethanol Concentration

As shown in Figure 1A, the yield of total flavonoids increased gradually as the ethanol concentration rose from 50% to 80%, reaching a peak of 11.50 mg/g at 80% ethanol. However, when the ethanol concentration exceeded 80%, the yield began to decline. This decrease may be attributed to the reduced affinity of high-concentration ethanol for the polar groups in flavonoid compounds, hindering their extraction from the plant. Additionally, higher ethanol concentrations may extract other fat-soluble impurities, further reducing the purity of the flavonoids [25]. The univariate analysis results indicated that the optimal extraction solvent was 80% ethanol.

#### 2.1.2. Extraction Temperature

As shown in Figure 1B, the yield of flavonoids increased significantly as the extraction temperature was raised from 60 °C to 90 °C. However, when the temperature exceeded 90 °C, the yield of flavonoids decreased sharply. The reason for this result may be that the high extraction temperature accelerates the molecular motion and increases the intermolecular collisions, thus contributing to the oxidation and degradation of the flavonoid components [26]. The results suggest that the optimal extraction temperature is 90 °C.

#### 2.1.3. Solid-to-Liquid Ratio

Figure 1C illustrates that the flavonoid yield from *Polygonum perfoliatum* L. initially increased and then decreased as the solid-to-liquid ratio increased. When the ratio was raised from 1:10 to 1:30 g/mL, the yield of flavonoids gradually rose, peaking at 1:30 with a maximum yield of 13.22 mg/g. However, beyond a 1:30 ratio, the yield began to decline, likely because an excessive amount of extraction solvent may extract more impurities while diluting the flavonoid concentration [27]. In addition, a high material-to-liquid ratio means excessive solvent consumption, which not only increases costs but also affects the environment. Therefore, the optimal solid-to-liquid ratio is 1:30 g/mL.

#### 2.1.4. Extraction Time

As shown in Figure 1D, the yield of flavonoids increased with longer extraction times from 0.5 to 1.5 h, reaching a peak yield of 14.05 mg/g at 1.5 h. However, when the extraction time exceeded 2.5 h, a decline in flavonoid yield was observed, likely due to the degradation of flavonoids caused by prolonged heating [28], which suggests that increasing the extraction time can improve the extraction efficiency of flavonoids within a certain time range. Between 1.5 and 2.5 h, the yield remained stable. Considering the trade-off between maximizing yield and minimizing energy consumption, the optimal extraction time was determined to be 1.5 h.

#### 2.1.5. Number of Extractions

Figure 1E illustrates the effect of varying the number of extractions on flavonoid yield. The yield peaked at 14.39 mg/g when the number of extractions was increased to two. Therefore, the optimal number of extractions was determined to be two.

### 2.2. Response Surface Optimization Results for Flavonoid Extraction

#### 2.2.1. Model Fit and Significance Test

The response surface test data (Table 1) were analyzed using Design Expert 8.0.6 software, yielding the following multiple quadratic regression equation for flavonoid yield (*Y*) in relation to factors *A*, *B*, and *C*: *Y* = −49.57042 + 0.55575 × *A* + 0.94370 × *B* − 0.074033 × *C* + 0.00420600 × *A* * *B* + 0.00292850 × *A* × 30 + 0.00482392 × *B* × 30 − 0.00628942 × *A*^2^ − 0.00801134 × *B*^2^ − 0.00917984 × *C*^2^. Analysis of variance (ANOVA) for the regression model (Table 2) yielded a *p*-value < 0.0001 (*p* < 0.01), indicating that the model is statistically significant. The *p*-value for the lack-of-fit term was 0.1946 (*p* > 0.05), suggesting that the model fits well. The predicted coefficient of determination (R^2^_Pred_) was 0.8433, which is close to the adjusted coefficient of determination (R^2^_Adj_) of 0.9675, demonstrating a high degree of agreement between the predicted and experimental values. Additionally, the coefficient of variation (C.V.) was 1.21% (<10%), confirming the reliability and stability of the experimental results.

#### 2.2.2. Model Verification

Based on the analysis of the results in Table 1 and Table 2, as well as Figure 2, the optimal process conditions for extracting total flavonoids from *Polygonum perfoliatum* L. were determined to be as follows: an ethanol concentration of 82.00%, a temperature of 90.29 °C, and a solid-to-liquid ratio of 1:32.78 (g/mL). Under these conditions, the flavonoid yield in three parallel experiments was 14.98 ± 0.11 mg/g, which showed no significant difference from the predicted yield of 14.61 mg/g. This confirmed the accuracy and reliability of the model. The results of this study establish the optimal process parameters for the crude flavonoid extract, providing a solid foundation for the subsequent purification of flavonoids.

### 2.3. Purification of Total Flavonoids Using Macroporous Resin

#### 2.3.1. Screening of Macroporous Resin Models

The adsorption and desorption performance of macroporous adsorption resins is influenced by factors such as the chemical structure, polarity, surface area, and pore size of the components [29,30]. To enhance the extraction of flavonoids from *Polygonum perfoliatum* L., seven macroporous resins with varying polarities were selected for investigation in this experiment (Table 3). The adsorption capacity of these resins for the total flavonoids was ranked as follows: HPD-600 > D-101 > AB-8 > HP-20 > HPD-100 > NKA-9 > DM130. In terms of desorption capacity, the ranking was as follows: D-101 > HP-20 > AB-8 > HPD-100 > NKA-9 > DM130 > HPD-600. By comprehensively comparing the static adsorption and desorption capacities of these seven resins, the non-polar D-101 macroporous resin emerged as the most effective. Consequently, D-101 was selected for the follow-up study.

#### 2.3.2. Static Adsorption and Desorption Kinetic Curves of D101 Macroporous Resin

The adsorption and desorption kinetic curves for the total flavonoids of *Polygonum perfoliatum* L. using D101 macroporous resin are shown in Figure 3. From Figure 3A, it is evident that the resin adsorbed flavonoids rapidly within the first 1 to 3 h, slowed down between 3 and 4 h, and reached saturation at 4 h. Figure 3B indicates that the D101 resin released flavonoids rapidly in the first 2 h and reached equilibrium thereafter. In summary, the optimal adsorption time for the D101 resin was 4 h, while the optimal desorption time was 2 h.

#### 2.3.3. Results of One-Factor Adsorption Experiments

The effect of different pH values of the drug solution on the adsorption performance of total flavonoids is shown in Figure 4A. The adsorption performance exhibited a trend of increasing and then decreasing with the rise in pH, with the best adsorption observed at pH 3. A pH that is too low may cause the flavonoids to form salts, while a pH that is too high may reduce the flavonoids’ ability to form hydrogen bonds with the resin, thereby decreasing adsorption performance; the optimum pH for the purification of *Hippophae rhamnoides* L. flavonoids to obtain the sample solution was 5.0, which proves that flavonoids are more conducive to the adsorption of the resin under acidic conditions, in line with the experimental results [31]. In addition, the different types of flavonoid compounds from different sources and the type of resin used may account for the differences in optimal pH. However, the *Polygonum perfoliatum* L. extract itself has a pH of 3.85, which is likely due to the fact that the flavonoids in the extract primarily contain phenolic hydroxyl groups in their structure. Therefore, a pH of 3 was selected for the sample to achieve optimal adsorption performance.

With an increase in the diameter-to-height ratio, the adsorption rate exhibited an upward trend, whereas the specific adsorption amount showed a downward trend (Figure 4B). When the diameter-to-height ratio was 1:12, the adsorption rate was at its maximum. However, due to the excessive height of the resin bed, the penetration of the upper sample solution was poor, leaving the resin at the lower end of the column unused (still white after adsorption), resulting in resin waste. Comparatively, the adsorption rates for diameter-to-height ratios of 1:10 and 1:8 were similar, but the specific adsorption capacity decreased slightly. After a comprehensive evaluation, the optimal diameter-to-height ratio was determined to be 1:8.

Excessively small sample volumes during the purification process lead to low resin utilization, resulting in waste and prolonged experimental periods [32]. Conversely, excessively large sample volumes, due to the limited internal space of the resin microspheres, hinder complete adsorption of flavonoids and reduce purification efficiency [33]. As shown by the percolation curve in Figure 4C, the mass concentration of total flavonoids in the third effluent reached one-tenth of the concentration in the initial sample, marking the leakage point [34]. Based on this, an optimal sample volume of 1.4 BV (30 mL) was selected.

Figure 4D illustrates the effect of sample concentration on the adsorption performance of flavonoids. As the sample concentration increased, the adsorption rate rose, while the specific adsorption amount declined. This trend may be attributed to the greater total amount of flavonoids coming into contact with the resin per unit surface area at higher concentrations, leading to a higher adsorption rate. However, it may also cause sample leakage without adsorption, resulting in a reduction in specific adsorption capacity [30,35]. To balance the specific adsorption capacity and adsorption rate, an optimal sample concentration of 0.6 g/mL was chosen.

Figure 4E highlights the impact of sample flow rate on the adsorption performance of D101 resin. As the flow rate increased, the adsorption capacity of the resin decreased, likely due to insufficient contact time between the flavonoid components and the macroporous resin at higher flow rates [36]. Within the range of 0.5 to 1.0 mL/min, the variation in flow rate had a minimal impact on both the specific adsorption capacity and the adsorption rate. Thus, a sample flow rate of 1.0 mL/min was selected to shorten the experimental duration.

#### 2.3.4. Results of One-Factor Elution Experiments

From Figure 5A, it can be observed that the flavonoid concentration in the aqueous wash was lowest when the amount of distilled water reached 0.95 BV (20 mL). Beyond this point, the flavonoid concentration in the wash exhibited an increase followed by a decrease with the rising volume of distilled water. It was hypothesized that flavonoid glycosides adsorbed into the resin column might have been eluted as the water volume increased. Therefore, the optimal amount of distilled water was determined to be 0.95 BV.

Figure 5B illustrates that when the eluent volume reached 70 mL, no flavonoid components were detected in the additional eluent, indicating that the flavonoids adsorbed on the resin had been completely eluted. Thus, the optimal eluent dosage was established as 3.3 BV.

The effect of different ethanol concentrations on the desorption capacity was studied, as shown in Figure 5C. The desorption capacity increased initially and then decreased with rising ethanol concentration, peaking at 60%. It was speculated that lower ethanol concentrations might be insufficient to effectively compete for adsorption sites on the resin, resulting in inadequate desorption. Conversely, excessively high ethanol concentrations could reduce the solubility of flavonoids, leading to incomplete elution [37]. Therefore, 60% ethanol was selected as the eluent.

Figure 5D shows the effect of different elution flow rates on the desorption capacity of flavonoids. The desorption capacity increased at first and then decreased as the flow rate rose, with the best desorption effect achieved at a flow rate of 1.0 mL/min. Consequently, 1.0 mL/min was chosen as the optimal elution flow rate.

#### 2.3.5. Optimal Condition Verification

The optimal conditions for adsorption and desorption were determined through one-way experiments. D101 macroporous resin was selected to fill the chromatographic column, with a diameter-to-height ratio of 1:8. The sample solution volume was set at 1.5 BV, with a pH of 3, a concentration of 0.6 g/mL, and a flow rate of 1.0 mL/min. Unadsorbed flavonoids were removed by eluting with 0.95 BV of distilled water, followed by elution with 3.3 BV of 60% ethanol at a flow rate of 1.0 mL/min. The eluate was collected and freeze-dried to calculate the flavonoid purity. The experiment was repeated three times under these conditions, yielding a flavonoid purity of 43.00 ± 2.55% in *Polygonum perfoliatum* L., which was 3.38 times higher than that of the crude extract (12.74 ± 1.04%).

### 2.4. Purification of Flavonoids by Coupling Macroporous Resins with *Polyamides*

#### 2.4.1. Screening of Polyamide Mesh

The mesh size of polyamides significantly affects the purification efficiency, as shown in Figure 6A. Smaller polyamide particle sizes resulted in poorer adsorption and desorption of flavonoids from *Polygonum perfoliatum* L. Among the tested sizes, polyamides with a pore size of 14–30 mesh exhibited the best adsorption and elution capacities for flavonoids. This could be attributed to the fact that excessively small pore sizes may cause clogging, hindering the diffusion and adsorption of flavonoid molecules [38]. Therefore, 14–30 mesh polyamides were selected for the subsequent purification of total flavonoids from *Polygonum perfoliatum* L.

#### 2.4.2. Comparison of the Order of Macroporous Resins in Combination with Polyamides

To enhance enrichment, purification, and efficiency, chromatographic columns with two mixed packing materials were employed. Li et al. showed that the purity of flavonoids in *Zostera marina* L. was significantly increased from (14.97 ± 0.51)% to (71.14 ± 0.79)% by placing the sample solution in the upper part of a macroporous resin column, followed by adsorption onto polyamide and elution with 70% ethanol. However, this method is relatively complicated and requires the addition of a spacer to separate the two different fillers, which is not conducive to the subsequent process. [39]. Similarly, Jin et al. achieved highly efficient flavonoid enrichment by sequentially purifying Ampelopsis grossedentata extract with polyamide and macroporous resin, increasing the purity from 55.00% to 80.75% with extended experimental cycles [40]. The combination order of polyamide and macroporous resin significantly impacts the purification outcome. To investigate this, three packing arrangements were tested: (1) macroporous resin in the upper layer and polyamide in the lower layer, separated by filter paper; (2) polyamide in the upper layer and macroporous resin in the lower layer, separated by filter paper; and (3) auniform mixture of polyamide and macroporous resin before column packing.

As shown in Figure 6B, the adsorption rates for these configurations were 89.74%, 88.98%, and 89.36%, respectively, while the desorption rates were 88.48%, 85.27%, and 89.43%, respectively. The flavonoid purities achieved were 51.35%, 47.43%, and 50.56%, respectively. Both the first and third configurations showed superior purification performance compared to the second. Due to its operational simplicity and suitability for industrial-scale production, the third configuration, where polyamide and macroporous resin are mixed before column loading, was selected for subsequent flavonoid purification. It is hypothesized that polyamide and macroporous resin exhibit different adsorption mechanisms. Polyamide primarily adsorbs flavonoids through hydrogen bonding, while macroporous resin utilizes physical adsorption and molecular sieving. By combining these materials, their complementary mechanisms synergize, maximizing adsorption efficiency and purification performance. This approach provides a more effective method for enriching and purifying flavonoids.

#### 2.4.3. Ratio of Macroporous Resin to Polyamide

As shown in Figure 6C, the effect of the dosage ratio of macroporous resin to polyamide was evaluated. By considering the adsorption rate, desorption rate, and purity, optimal flavonoid purification was achieved at a dosage ratio of 1:9. This ratio likely represents an ideal balance between physical adsorption, molecular sieve action, and hydrogen bonding adsorption, enabling highly selective enrichment of flavonoids [41]. Therefore, the 1:9 ratio was determined to be the optimal dosage for both materials.

#### 2.4.4. Static Adsorption and Desorption Kinetic Curves for Mixed Packing Materials

The static adsorption and desorption kinetics curves revealed the adsorption and desorption characteristics of flavonoids in the mixed packing chromatography columns, as shown in Figure 7. From Figure 7A, it is evident that the adsorption rate increased rapidly during the first 3 h, slowed between 3 and 4 h, and reached equilibrium at 4 h. The adsorption rate of flavonoids in the mixed packing columns was notably high. The initial rapid adsorption may be attributed to the abundant availability of adsorption sites within the mixed packing, which facilitated a faster rate of adsorption; when a certain time is reached, most of the adsorption sites are occupied, and the adsorption rate thus slows down until it reaches equilibrium [42]. Figure 7B illustrates that the desorption rate of flavonoids was faster during the first 3 h and reached equilibrium after approximately 3 h. The higher desorption rate in the initial phase is likely due to the sufficient driving force for desorption early in the process. Based on these findings, the optimal adsorption time was determined to be 4 h, while the optimal desorption time was set at 3 h.

#### 2.4.5. Results of One-Way Experiments

Figure 8A illustrates the sample volume of the mixed column containing macroporous resin and polyamide. Due to the column’s limited adsorption capacity, the amounts of flavonoids in the effluent increased as the sample volume rose. When the total flavonoid concentration in the fifth column pass reached one-tenth of the concentration in the feed solution, the leakage point was identified [34]. Accordingly, a sample volume of 50 mL (2.4 BV) was selected as optimal.

As shown in Figure 8B, the adsorption capacity of the hybrid column for flavonoids initially increased and then decreased with rising sample concentrations. This trend may result from the availability of sufficient adsorption sites at lower sample concentrations, which facilitated higher flavonoid adsorption. However, as the sample concentration increased, competition for the limited binding sites from impurities likely reduced the adsorption capacity [43]. The peak adsorption performance was observed at a sample concentration of 0.8 g/mL, establishing it as the optimal concentration.

To remove unadsorbed flavonoids and impurities from the mixed column, the volume of distilled water was evaluated, as shown in Figure 8C. The flavonoid concentration in the aqueous wash decreased with increasing water dosage, stabilizing at 90 mL. Therefore, 4.3 BV (90 mL) was determined to be the optimal distilled water dosage.

Figure 8D shows that the desorption rate initially increased and then decreased with rising ethanol concentration. The maximum desorption rate was achieved with 80% ethanol, making it the optimal eluent concentration for the hybrid column. As depicted in Figure 8E, no flavonoids were detected in the eluate when the elution volume reached 90 mL, indicating complete elution from the hybrid column. Thus, 4.3 BV (90 mL) was established as the optimal elution volume.

#### 2.4.6. Verification of Optimal Conditions

The pretreated macroporous resin and polyamide were mixed in a 1:9 ratio and loaded into the chromatographic column slowly and wet-packed, achieving a column height-to-diameter ratio of 1:8. The sample solution, with a concentration of 0.8 g/mL, was introduced at a volume of 2.4 BV. Decontamination was performed using 4.3 BV of distilled water at a flow rate of 1.0 mL/min, followed by elution with 4.3 BV of 80% ethanol solution. The eluate was collected and freeze-dried, and the flavonoid purity was determined through three parallel tests. After purification using the mixed column of macroporous resin and polyamide, the flavonoid purity reached 59.02 ± 2.23%, representing a 1.37-fold improvement compared to macroporous resin alone (43.00 ± 2.55%) and a 4.63-fold increase relative to the crude extract (12.74 ± 1.04%). This study marks the first successful enrichment and purification of flavonoids from *Polygonum perfoliatum* L. using a combination of macroporous resin and polyamide fillers. Compared to the use of a single packing material, the hybrid column efficiently separated various components, yielding a higher-purity target compound. Jin et al. separated and purified the extract accordingly over polyamide and macroporous resin [40], achieving only a 1.45-fold increase in purity, and the method was cumbersome and prolonged the experimental period; conversely the purity was increased by 4.63-fold after enrichment using a hybrid chromatographic column with polyamide and macroporous resin. By integrating the two materials into a single column, our approach simplifies operation, enhances purification efficiency, and offers a scalable method suitable for subsequent industrial production.

### 2.5. Analysis of Flavonoid Compounds Using HPLC-MS/MS

The purified flavonoids were analyzed and identified using HPLC-MS/MS. The compositional identification of the purified flavonoids from *Polygonum perfoliatum* L. was based on secondary mass spectral data, leveraging databases such as Thermo mzCloud and Thermo mzVault. The total ion chromatograms of the purified flavonoids in both positive and negative ion modes are shown in Figure 9. A total of 79 flavonoids were identified from *Polygonum perfoliatum* L., with most existing either in their free form or as glycosides. These included 9 isoflavonoids, 7 dihydroflavonoids, 36 flavonols, 4 flavanols, and 23 other flavonoids. Detailed *HPLC-MS* detection data are provided in Appendix A. In contrast, Liu et al. identified only 26 flavonoids from *Polygonum perfoliatum* L. [1], underscoring the more extensive profiling achieved in this study.

### 2.6. Total Flavonoid Anti-VZV Assay

#### 2.6.1. Measurement of TCID50

As shown in Table 4, the cellular morphology post-infection with VZV was assessed using the cytopathic effect (CPE) method, and the viral titer (TCID50) was determined using the Reed–Muench method. The experiment was repeated three times to ensure accuracy, yielding a TCID50 of 10^−5.46^ for this varicella-zoster virus.

#### 2.6.2. Determination of Maximum Non-Toxic Concentration

It is widely accepted that a compound is considered non-significantly toxic to cells if its absorbance ratio (absorbance of treated cells/absorbance of positive control) is ≥70% [32]. As shown in Figure 10, the maximum non-toxic concentrations of acyclovir, purified flavonoids, and fresh juice were determined to be 500 μg/mL, 250 μg/mL, and 250 μg/mL, respectively.

#### 2.6.3. Anti-VZV Activity

The anti-VZV activity of different dose groups of each drug is shown in Figure 11. When the mass concentration was 31.3 μg/mL, the cell survival rate was 66.32% in the acyclovir group, 43.28% in the purified flavonoid extract group, and 39.58% in the fresh juice group. A highly significant difference in anti-VZV activity was observed between the purified flavonoid extract and the fresh juice group (*p* < 0.001). The IC_50_ values were 8.067 μg/mL for acyclovir, 40.10 μg/mL for the purified flavonoid extract, and 56.30 μg/mL for the fresh juice of *Polygonum perfoliatum* L. Both the purified flavonoid extract and the fresh juice of *Polygonum perfoliatum* L. demonstrated good anti-VZV effects. Traditionally, fresh *Polygonum perfoliatum* L. was pounded and applied to the affected area to effectively treat herpes zoster [1]. In this study, the crude extract of *Polygonum perfoliatum* L. was purified and compared with the traditional fresh juice method for anti-VZV activity. The results indicated that the purified flavonoid extract exhibited stronger antiviral activity than the fresh juice.

### 2.7. Examination of the Antioxidant Activity of Flavonoids of Polygonum perfoliatum L.

#### 2.7.1. Determination of Total Reducing Power of Total Flavonoids

For the determination of total reducing power, a higher absorbance value indicates stronger reducing power of the sample [44]. As shown in Figure 12A, at a mass concentration of 0.160 mg/mL, the absorbance values were as follows: 2.840 for vitamin C, 0.790 for the purified flavonoid extract, 0.410 for the crude extract, and 0.170 for the fresh juice. The reducing power followed the order: vitamin C > purified flavonoid extract > crude extract > fresh *Polygonum perfoliatum* L. juice. This trend may be attributed to the increased flavonoid content after purification, which enhances antioxidant capacity. Additionally, the antioxidant activity of the total flavonoids from *Polygonum perfoliatum* L. reached saturation at a reducing capacity corresponding to a concentration of 0.600 mg/mL.

#### 2.7.2. Determination of the Scavenging Capacity of Total Flavonoids Against DPPH Free Radicals

The antioxidant activities of the samples were evaluated based on their DPPH radical scavenging rates, as shown in Figure 12B. At a mass concentration of 0.12 mg/mL, the scavenging rates were 97.220% for vitamin C, 95.360% for the purified flavonoid extract, 86.260% for the crude extract, and 15.000% for the fresh juice. The IC_50_ values were 0.009 mg/mL for vitamin C, 0.016 mg/mL for the purified flavonoid extract, 0.041 mg/mL for the crude extract, and 0.810 mg/mL for the fresh juice. The order of scavenging ability for DPPH radicals was as follows: vitamin C > purified flavonoid extract > crude extract > fresh juice. The significant improvement in DPPH radical scavenging ability after purification highlights the enhanced efficacy of the total flavonoids in the purified extract of *Polygonum perfoliatum* L.

#### 2.7.3. Determination of Free Radical Scavenging Capacity of Total Flavonoids (ABTS)

The ABTS radical scavenging capacities of the samples are illustrated in Figure 12C. The scavenging ability increased with concentration, reaching equilibrium within the range of 0.020–0.900 mg/mL. At a concentration of 0.100 mg/mL, the ABTS radical scavenging rates were 99.530% for vitamin C, 93.060% for the purified flavonoid extract, 54.540% for the crude extract, and 9.960% for the fresh juice. The IC_50_ values were 0.011 mg/mL for vitamin C, 0.013 mg/mL for the purified flavonoid extract, 0.028 mg/mL for the crude extract, and 0.626 mg/mL for the fresh juice. The antioxidant capacity rankings were as follows: vitamin C > purified flavonoid extract > crude extract > fresh juice. In conclusion, these results demonstrate that the total flavonoids of *Polygonum perfoliatum* L. exhibit strong scavenging effects on both DPPH and ABTS radicals, with a significant enhancement in activity following purification.

## 3. Materials and Methods

### 3.1. Materials and Chemicals

The *Polygonum perfoliatum* L. herb (batch number: 230501) was purchased from Guiyang Daosheng Health Industry Co., Ltd. (Guiyang, China), and authenticated by Associate Professor Yuan Ye, School of Pharmacy, Guizhou University of Traditional Chinese Medicine. Rutin (batch number: D13HB202516, purity ≥ 98%) was purchased from Shanghai Yuanye Biotechnology Co., Ltd. (Shanghai, China), 95% analytically pure (batch number: 20230424) was purchased from Chongqing Chuandong Group Co., Ltd. (Chongqing, China), vitamin C (batch number: PS020110, purity 99.73%) was purchased from Chengdu Pusi Bio-Science and Technology Co., Ltd. (Chengdu, China), and the macroporous adsorption resins (HPD-600, NKA-9, DM130, AB-8, D-101, HPD-100, HP-20) and *polyamide* (14–30 mesh, 30–60 mesh, 60–100 mesh, 100–200 mesh) were purchased from Beijing Solebo Technology Co. (Beijing, China). AlCl_3_ reagent was purchased from Tianjin Zhiyuan Chemical Reagent Co. (Tianjin, China). Varicella zoster virus (ATCC VR-1433™) was purchased from ATCC, USA. (Manassas, VA, USA).

### 3.2. Determination of Total Flavonoids

Precisely 10.15 mg of rutin was weighed as the control product and dissolved in 70% ethanol in a volumetric flask to prepare a control solution with a concentration of 0.203 mg/mL. Measures of 0 mL, 0.4 mL, 0.5 mL, 0.7 mL, 0.8 mL, 1 mL, 1.2 mL, and 1.3 mL of the control solution were pipetted into separate 10 mL volumetric flasks. An AlCl_3_ solution was added for color development, and the solutions were then mixed thoroughly and allowed to stand for 20 min [45]. The absorbance was measured at 405 nm, using the corresponding reagents as blanks. By plotting the mass concentration of rutin (*X*) on the horizontal axis and the absorbance (*Y*) on the vertical axis, the standard curve *Y* = 28.95 *X*-0.0064, *r* = 0.9994 was obtained. The flavonoid yield was calculated as follows:(1)$$\mathrm{Flavonoid}\;\mathrm{yield}\;(\mathrm{mg}/g)=\frac{C*V}{M}$$

*C* is the concentration of the extract (mg/mL); *V* is the volume of the extract (mL); *M* is the mass of the powder of *Polygonum perfoliatum* L. (g).

### 3.3. One-Way Experimental Design for Flavonoid Extraction

The effects of various factors on the flavonoid yield were investigated, including the ethanol concentration (50%, 60%, 70%, 80%, and 90%), extraction temperature (60 °C, 70 °C, 80 °C, 90 °C, and 100 °C), extraction time (0.5, 1, 1.5, 2, 2.5, and 3 h), solid-to-liquid ratio (1:5, 1:10, 1:20, 1:30, and 1:40 g/mL), and number of extractions (1, 2, 3, and 4).

### 3.4. Response Surface Methodology (RSM) Experiments

To further optimize the flavonoid yield, the effects of the ethanol concentration, extraction temperature, and solid-to-liquid ratio were analyzed using a three-factor, three-level response surface design. The optimization was performed with Design-Expert V8.0.6.1 software, building on one-factor experiments. The specific levels and factors used in the RSM design are detailed in Table 5.

### 3.5. Purification of Flavonoids Using Macroporous Resins

#### 3.5.1. Selection of Large-Pore Adsorbent Resin Types

The adsorption performance and efficiency of the resin are usually influenced by factors such as polarity, pore size, and specific surface area, and the physical parameters of different models of macroporous adsorption resins are shown in Table 6. The static adsorption method was employed to select the preferred type of macroporous adsorbent resin. Approximately 1 g (wet mass) of each pre-treated macroporous adsorbent resin— AB-8, D-101, HPD100, HPD600, HP20, NKA-9, and DM130—was taken. To each type, 20 mL of a 0.45 g/mL extract of *Polygonum perfoliatum* L. was added. The mixtures were placed in a constant-temperature shaking water bath at room temperature (25 °C) and shaken at 100 rpm for 24 h. Afterward, the mixtures were filtered, and the absorbance of the filtrate was measured. The resins were washed twice with 50 mL of distilled water and filtered. For desorption, precisely 20 mL of 70% ethanol was added, and the mixtures were placed back into the constant-temperature shaking water bath for 24 h (25 °C, 100 rpm). After complete desorption, the solutions were filtered to obtain the desorption solutions. An appropriate amount of each desorption solution was precisely taken, and its absorbance was measured. The adsorption parameters were calculated using the following formula [46]:(2)$$\mathrm{Specific}\;\mathrm{adsorption}\;\mathrm{capacity}\;(\mathrm{mg}/g)=\frac{(C_{0}-C_{1})*V}{W}$$
(3)$$\mathrm{Adsorption}\;\mathrm{rate}\;(\%)=\frac{C_{0}-C_{1}}{C_{0}}*100\%$$
(4)$$\mathrm{Specific}\;\mathrm{desorption}\;\mathrm{capacity}\;(\mathrm{mg}/g)=\frac{C_{2}*V_{2}}{W}$$
(5)$$\mathrm{Desorption}\;\mathrm{rate}\;(\%)=\frac{C_{2}*V_{2}}{(C_{0}-C_{1})*V_{1}}*100\%$$
(6)$$\mathrm{Fineness}\;(\%)=\frac{C_{2}*V_{2}*N}{M}*100\%$$

*C*_0_ is the initial solubility of the upper sample solution (mg/mL); *C*_1_ is the concentration of the equilibrium solution (mg/mL); *C*_2_ is the mass concentration of the desorbent solution (mg/mL); *V* is the volume of the upper sample solution (mL); *V*_1_ is the volume of the adsorbent solution (mL); *V*_2_ is the volume of the desorbent solution (mL); *W* is the weight of the resin (g); *M* is the weight of the purified material powder (mg); *N* is the dilution multiple.

#### 3.5.2. Kinetic Curves of Adsorption and Desorption

Macroporous resin (1g) was weighed, and 20 mL of the *Polygonum perfoliatum* L. extract was added. The mixture was shaken in a constant temperature oscillation water bath (100 rpm, 25 °C). At 1-h intervals, 500 μL of the supernatant was removed until adsorption equilibrium was reached. The concentration of total flavonoids in the supernatant was determined, and the static adsorption kinetic curve was plotted. After the adsorption process, 20 mL of 70% ethanol was added to the resin for desorption. At 1-h intervals, 500 μL of the supernatant was collected until desorption equilibrium was reached. The concentration of total flavonoids in the desorption solution was measured, and the static desorption kinetic curve was plotted.

#### 3.5.3. Effect of pH on Static Adsorption

Approximately 1 g (wet mass) of pretreated D101-type macroporous adsorbent resin was accurately weighed and placed into a 100 mL stoppered conical flask. Subsequently, 20 mL of an extract solution with a mass concentration of 0.45 g/mL was added. The pH of the solution was adjusted to 2, 3, 3.85 (drug solution), 4, 5, 6, 7, or 8 using 1 mol/L of hydrochloric acid and 1 mol/L of sodium hydroxide. The flask was then shaken at 100 rpm for 24 h in a water bath set at 25 °C. After the adsorption process, the total flavonoid concentration in the adsorbent was measured, and the specific adsorption amount and adsorption rate were calculated.

#### 3.5.4. One-Factor Experiments for the Purification of Flavonoid Compounds Using Macroporous Resins

Approximately 7.8 g, 11.8 g, 16.5 g, 20.5 g, and 23.8 g (wet mass) samples of pretreated macroporous adsorbent resin were precisely weighed, corresponding to chromatography column diameter-to-height ratios of 1:4, 1:6, 1:8, 1:10, and 1:12, respectively. Using the wet packing method, the resins were slowly loaded into chromatography columns (1.5 cm × 30 cm). Solutions were extracted at different concentrations (0.3, 0.4, 0.45, 0.5, 0.6, 0.7, and 0.8 g/mL, equivalent to the amount of raw drug), and a pH of 3 was maintained in the columns of varying volumes. The solutions were passed through the columns at flow rates of 0.5, 1, 1.5, 2, and 3 mL/min, and the effluent was collected to determine absorbance. Based on the results, the optimal sample concentration and flow rate were determined. To remove unadsorbed flavonoids and impurities from the resin prior to desorption, different column volumes of water were used, and the optimal water washing volume was established. For desorption, ethanol solutions with concentrations ranging from 50% to 90% were passed through the columns at varying flow rates (1, 1.5, 2, and 3 mL/min). Elution curves were plotted to determine the optimal ethanol concentration and flow rate. Finally, the desorbed solutions were collected, freeze-dried, and analyzed to determine their purity.

### 3.6. Purification of Flavonoids by Combining Macroporous Resin and Polyamide

#### 3.6.1. Screening of Polyamide Mesh

The static adsorption and desorption rates of total flavonoids were used as evaluation indices. Polyamides with different mesh sizes—14–30 mesh, 30–60 mesh, 60–100 mesh, and 100–200 mesh—were selected based on the static adsorption method.

#### 3.6.2. Order of Macroporous Resins in Combination with Polyamides

Seven grams of each pretreated macroporous resin (MAR) and polyamide (PA) (dosage ratio 1:1, diameter-to-height ratio 1:8) were precisely weighed and loaded into the column in varying orders. A sample solution containing 0.6 g/mL of biopharmaceuticals at pH 3 was added to the column at a flow rate of 1.0 mL/min, with a volume of 3.3 BV (70 mL). The column was then washed with 1 BV (21 mL) of water to remove impurities, followed by elution using 3.8 BV (80 mL) of 70% ethanol at a flow rate of 1.0 mL/min. The eluate was collected, and its absorbance at 405 nm was measured to determine the optimal loading order of MAR and PA.

#### 3.6.3. Dosage Ratio of Macroporous Resin to Polyamide

Pre-treated macroporous resin and polyamide were used to fill the chromatographic column at varying dosage ratios (0:10, 1:9, 2:8, 3:7, 4:6, 5:5, 6:4, 7:3, 8:2, 9:1, 10:0). A sample solution with a volume of 3.3 BV (70 mL) and a concentration of 0.6 g/mL was then added to the column. The column was washed with 1 BV (21 mL) of water to remove impurities, followed by elution with 3.8 BV (80 mL) of 70% ethanol. The adsorption rate, desorption rate, and flavonoid purity were determined for each dosage ratio.

#### 3.6.4. Adsorption and Desorption Kinetic Curves

A total of 1 g of mixed packing material (polyamide/macroporous resin = 9:1) was weighed and added to 20 mL of extraction solution with a concentration of 0.6 g/mL. The mixture was shaken in a thermostatic shaking bath at 100 rpm and 25 °C. Every hour, 500 μL of the supernatant was sampled until equilibrium was reached. The concentration of total flavonoids in the supernatant was measured, and an adsorption kinetic curve was plotted. After the adsorption process, 20 mL of 70% ethanol was added to the mixture, and desorption was conducted under the same conditions. The static adsorption kinetic curve was then plotted.

#### 3.6.5. One-Factor Investigation of Flavonoids Purified from Mixed Packing Materials

The pre-treated macroporous resin and polyamide were packed into the chromatographic column at a 1:9 dosage ratio, with a fixed diameter-to-height ratio of 1:8. Sample solutions at pH 3 with varying concentrations and column volumes were added to the column at a flow rate of 1.0 mL/min to determine the optimal sample volume and concentration. Next, different volumes of distilled water were passed through the column to remove impurities and identify the optimal water rinse volume. For elution, different volumes of 50% to 90% ethanol were used. The eluate was collected, and absorbance was measured to determine the optimal elution concentration and volume.

### 3.7. HPLC-MS/MS Analysis

#### 3.7.1. Test Sample

An appropriate amount of well-mixed sample powder was placed in a 2 mL centrifuge tube, and 1 mL of 70% methanol solution along with 3 mm steel beads were added. The mixture was ground using an automatic sample grinder at 70 Hz for 3 min, then vortexed for 10 min to ensure thorough mixing. The mixed solution was centrifuged at 4 °C for 10 min at 12,000 rpm. The supernatant was then filtered through a 0.22 μm micropore filter membrane, and 100 μg/mL of the internal standard solution (2-chlorophenylalanine) was added to achieve a final concentration of 1 mg/L.

#### 3.7.2. Chromatographic Conditions

A Zorbax Eclipse C18 column (1.8 μm × 2.1 mm × 100 mm) was used for full-spectrum analysis. The separation conditions were as follows: column temperature of 30 °C and a flow rate of 0.3 mL/min. The mobile phases consisted of (A) 0.1% formic acid in water and (B) pure acetonitrile. The injection volume was 2 μL. The gradient elution procedure for the sample is outlined in Table 7.

#### 3.7.3. Mass Spectrometry Conditions

Mass spectrometry was performed using a Q-Exactive quadrupole Orbitrap mass spectrometer (Thermo Fisher Scientific, Waltham, MA, USA). The parameters were set as follows: positive-negative ion mode, spray voltage of 3.5 kV/−3 kV, sheath gas flow rate of 45 arb, auxiliary gas flow rate of 15 arb, capillary temperature of 330 °C, and auxiliary temperature of 325 °C. The samples were analyzed using one-stage full scan (Full Scan, *m*/*z* 100–1500) and data-dependent two-stage mass spectrometry (dd-MS2, TopN = 5). The resolution was set to 120,000 for primary MS and 60,000 for secondary MS. Normalized collision energy (NCE) levels were set at 12.5%, 25%, and 35% to generate MS/MS spectra.

### 3.8. Analysis of Anti-Varicella-Zoster Virus (VZV) Activity

#### 3.8.1. Cell Culture and Virus Resuscitation

When Vero cells reached 75–80% confluency, the complete medium in the culture plate was removed and replaced with complete medium containing 2% fetal bovine serum (FBS). In a biosafety cabinet, 1 mL of complete medium was used to dissolve the VZV virus dry powder. The solution was then transferred to the prepared cells, mixed thoroughly, and incubated at 37 °C with 5% CO_2_. The cells were monitored for cytopathic effects (CPE), and when lesions covered approximately 90% of the cell layer, the cells were harvested for viral passaging. In the biosafety cabinet, virus-infected cells with over 90% lesion coverage were resuspended in 2 mL of virus preservation solution, transferred to cryovials, and stored at −80 °C. To use the virus, the frozen sample was thawed and centrifuged at room temperature. The resulting supernatant was collected as the cell-free VZV virus suspension.

#### 3.8.2. Determination of Viral Half-Tissue Infectivity (TCID50)

Cell suspensions containing Vero cells at a density of 2 × 10^5^ cells/mL were inoculated into 96-well plates, with each well receiving a 10-fold dilution of the virus. Eight replicate wells were established for each dilution, along with a cell control group. The 96-well plates were incubated at 37 °C with 5% CO_2_ for culture. Cell observations were made daily to monitor the occurrence of lesions, including non-specific cell death phenomena such as natural shedding and virus-specific cytopathic effects (CPEs) like cell swelling [47], rounding, shrinkage, aggregation, multinucleated giant cell formation, and detachment. These observations were conducted over a period of 6–7 days, with data collection occurring every 3 to 4 days. TCID50 was calculated using the Reed–Muench method.

#### 3.8.3. Cytotoxicity Assay

The cell suspension was seeded into a culture plate at a density of approximately 8000 cells per well. After aspirating the medium from each well, 100 μL of medium containing varying concentrations of the test samples was added to the wells. A blank control group (medium only) and a cell control group (cells with medium) were included, with three replicate wells set up for each condition. The 96-well plates were incubated at 37 °C for 24 h. Subsequently, 10 μL of CCK-8 reagent was added to each well, and the plates were gently shaken to ensure thorough mixing before continuing incubation for an additional 4 h. The optical density (OD) at 450 nm was measured using a microplate reader to calculate the cell survival rate.

#### 3.8.4. Anti-VZV Test

The cell suspension was seeded into the culture plate, with cell density adjusted using a cell counting plate. Groups for the purified flavonoid and fresh *Polygonum perfoliatum* L. juice were established, with the experimental group concentrations set based on the maximum non-toxic concentration, aiming for approximately 8000 cells per well. The cells were cultured for 24 h. Concurrently, positive drug, viral control, and normal cell control groups were set up in the 96-well plates. After 24 h of incubation, the medium was discarded, and 100 μL of drug-free medium was added to each well. Then, 10 μL of CCK-8 reagent was introduced into each well, and the plates were gently shaken to avoid air bubbles. The plates were incubated for an additional 4 h. The optical density values at 450 nm were measured using a microplate reader, and the data were recorded.

### 3.9. Analysis of Antioxidant Activity

#### 3.9.1. Total Reduction Measurement

Vitamin C, purified flavonoid, crude extract, and fresh juice of *Polygonum perfoliatum* L. were lyophilized and dissolved in anhydrous ethanol to prepare solutions at concentrations of 0.10, 0.12, 0.14, 0.16, 0.18, 0.20, 0.3, 0.4, 0.5, 0.6, 0.7, 0.8, 0.9, and 1.0 mg/mL. A 1% potassium ferricyanide solution and 0.2 mol/L phosphate buffer (pH = 6.6) were added to each solution, mixed thoroughly, and incubated in a 50 °C water bath for 20 min. Subsequently, 10% trichloroacetic acid solution was added, then the solution was mixed and centrifuged. The supernatant was collected and reacted with 0.1% ferric chloride solution at room temperature for 15 min. The absorbance at 700 nm was measured using pure water as the blank [48]. Each measurement was performed in triplicate.

#### 3.9.2. DPPH Free Radical Scavenging

Vitamin C, purified flavonoid, crude extract, and fresh juice of *Polygonum perfoliatum* L. were lyophilized and dissolved in anhydrous ethanol to prepare solutions at concentrations of 0.008, 0.01, 0.02, 0.04, 0.06, 0.08, 0.10, 0.12, 0.14, 0.16, 0.18, 0.20, 0.3, 0.4, 0.5, 0.6, 0.7, 0.8, 0.9, and 1.0 mg/mL. Following the method described by Zhang et al. [49], 100 μL of each test solution at different concentrations was mixed with an equal volume of DPPH ethanol solution. Additionally, 100 μL of the test solution was mixed with 100 μL of anhydrous ethanol, and 100 μL of anhydrous ethanol was mixed with 100 μL of DPPH ethanol solution (three sets of each). The mixtures were then placed in a 96-well plate shaker for 1 min, followed by a 30-min incubation at room temperature, protected from light. The absorbance at 517 nm was measured for each sample.

#### 3.9.3. ABTS Free Radical Scavenging Rate

Vitamin C, purified flavonoid, crude extract, and fresh juice of *Polygonum perfoliatum* L. were lyophilized and dissolved in anhydrous ethanol to prepare solutions at concentrations of 0.008, 0.01, 0.02, 0.04, 0.06, 0.08, 0.10, 0.12, 0.14, 0.16, 0.18, 0.20, 0.3, 0.4, 0.5, 0.6, 0.7, and 0.8 mg/mL. Following the method described by Ayyanna et al. [50], 100 μL of each test solution at varying concentrations was mixed with an equal volume of ABTS free radical working solution. Similarly, 100 μL of each test solution was mixed with 100 μL of anhydrous ethanol, and 100 μL of anhydrous ethanol was mixed with 100 μL of ABTS working solution (three replicates for each condition). The mixtures were placed in a 96-well plate shaker for 1 min, followed by a 30-min incubation at room temperature, protected from light. The absorbance at 734 nm was then measured.

### 3.10. Methods of Analysis

All experiments were independently repeated three times, and data are presented as mean ± standard deviation. All data were statistically analyzed using IBM SPSS Statistics 26.0; *p* < 0.05 was considered significant, and *p* < 0.001 was considered highly significant.

## 4. Conclusions

In this study, we successfully optimized the extraction process of total flavonoids from *Polygonum perfoliatum* L. using a one-way test, response surface analysis, and a quadratic regression equation model, which significantly increased the yield of flavonoids. For the first time, a mixed chromatographic column of polyamide and macroporous resin was used to effectively enrich and purify the flavonoids in *Polygonum perfoliatum* L., and the purity of the purified flavonoids was 4.63 times higher than that of the crude extract. Seventy-nine flavonoids were identified using ultra-performance liquid chromatography-tandem high-resolution mass spectrometry (UPLC-HRMS), which provided theoretical support for the study of the chemical composition of *Polygonum perfoliatum* L. Meanwhile, we also found that the purified flavonoid exhibited significant bioactivities in anti-VZV and in vitro antioxidant assays, which is important for the development of novel anti-VZV drugs and antioxidants. In addition, these results provide a scientific basis for further research and clinical application of *Polygonum perfoliatum* L., emphasizing its potential application value in the field of medicine. In future studies, we will conduct in-depth research on flavonoids’ mechanisms of action to better explain their roles in anti-VZV drugs and antioxidants; in addition, we should continue to explore the pharmacological effects of flavonoids in *Polygonum perfoliatum* L., as well as their potential applications in clinical therapy, in order to realize the maximum medicinal value of *Polygonum perfoliatum* L.

## Figures and Tables

**Figure 1 molecules-30-00029-f001:**
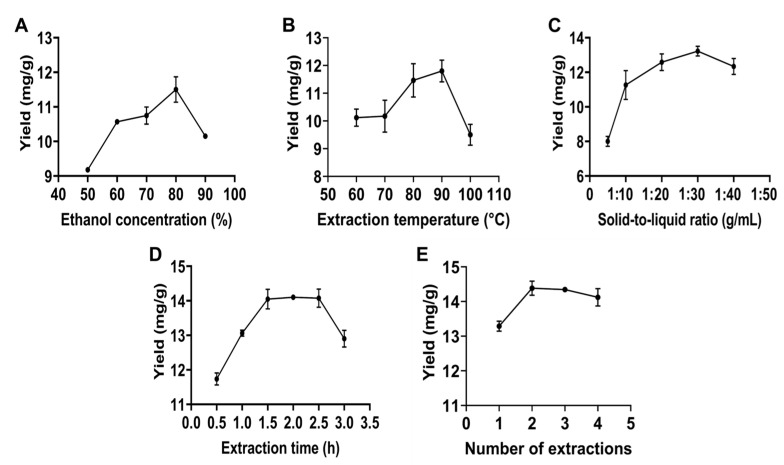
Effect of single factors on total flavonoid yield of *Polygonum perfoliatum* L. (**A**) Ethanol concentration; (**B**) extraction temperature; (**C**) solid-to-liquid ratio; (**D**) extraction time; (**E**) number of extractions.

**Figure 2 molecules-30-00029-f002:**
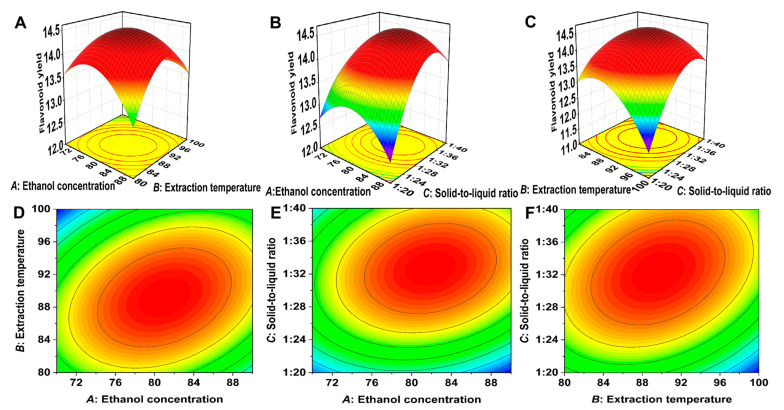
Response surface and contour plots between response variables in Box–Behnken. The 3D surface map (**A**) and the response surface contour diagram (**D**) for the interaction of ethanol concentration and temperature on the extraction efficiency of the total flavonoids from the *Polygonum perfoliatum* L. The 3D surface map (**B**) and the response surface contour diagram (**E**) for the interaction of ethanol concentration and solid-to-liquid on the extraction efficiency of the total flavonoids from the *Polygonum perfoliatum* L. The 3D surface map (**C**) and the response surface contour diagram (**F**) for the interaction of temperature and solid-to-liquid on the extraction efficiency of the total flavonoids from the *Polygonum perfoliatum* L.

**Figure 3 molecules-30-00029-f003:**
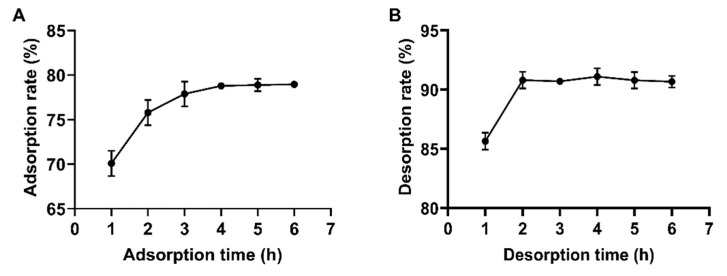
Kinetic profiles of static adsorption (**A**) and desorption (**B**) of total flavonoids of *Polygonum perfoliatum* L. on D101 macroporous resin.

**Figure 4 molecules-30-00029-f004:**
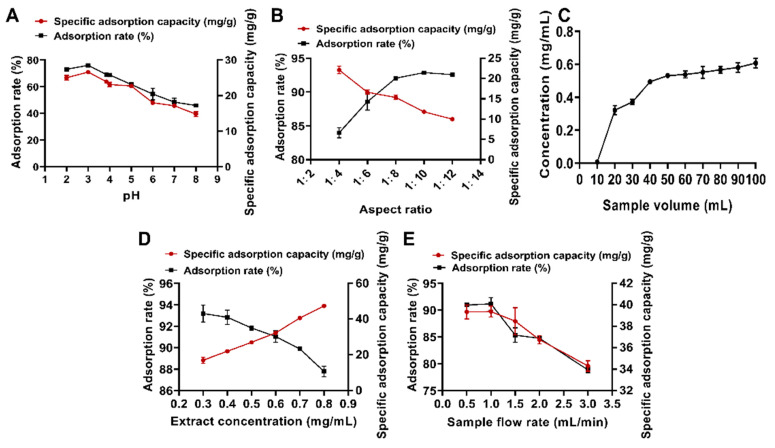
One-factor adsorption experiments with macroporous resins. (**A**) pH screening; (**B**) diameter-to-height ratio; (**C**) seepage curve; (**D**) sample concentration; (**E**) sample flow rate.

**Figure 5 molecules-30-00029-f005:**
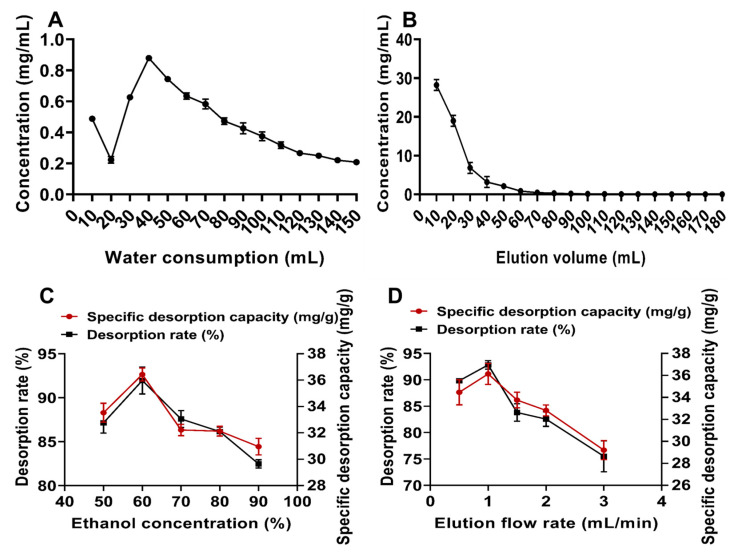
One-factor desorption experiment with macroporous resin. (**A**) Amount of aqueous wash; (**B**) elution profile; (**C**) ethanol concentration; (**D**) elution flow rate.

**Figure 6 molecules-30-00029-f006:**
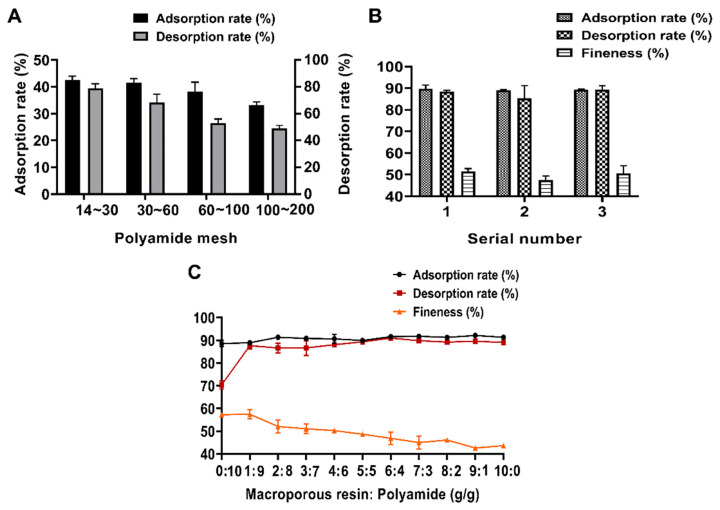
One-factor investigation of macroporous resins in combination with polyamides. (**A**) Polyamide mesh number; (**B**) comparison of the order of association; (**C**) dose ratio.

**Figure 7 molecules-30-00029-f007:**
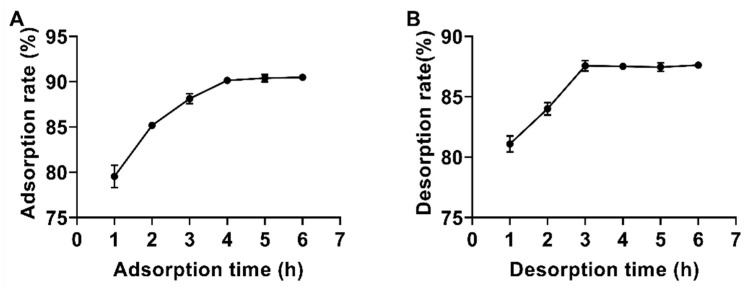
Kinetic curves of static adsorption (**A**) and desorption (**B**) of total flavonoids in mixed packing material for *Polygonum perfoliatum* L.

**Figure 8 molecules-30-00029-f008:**
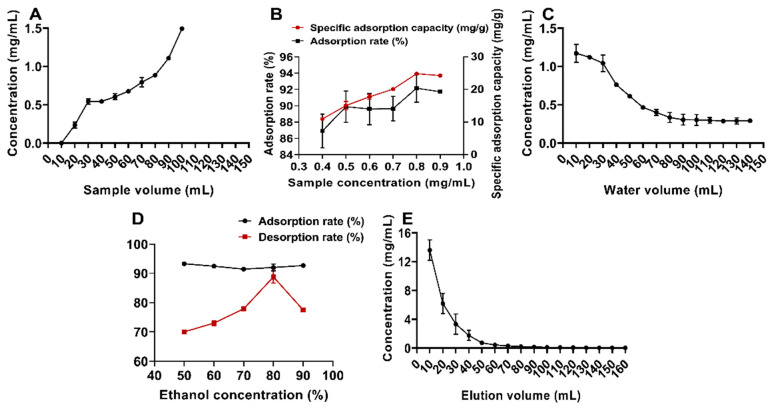
One-factor investigation of macroporous resins in combination with polyamides. (**A**) Percolation curve; (**B**) sample concentration; (**C**) water elution dosage; (**D**) elution concentration; (**E**) elution dosage.

**Figure 9 molecules-30-00029-f009:**
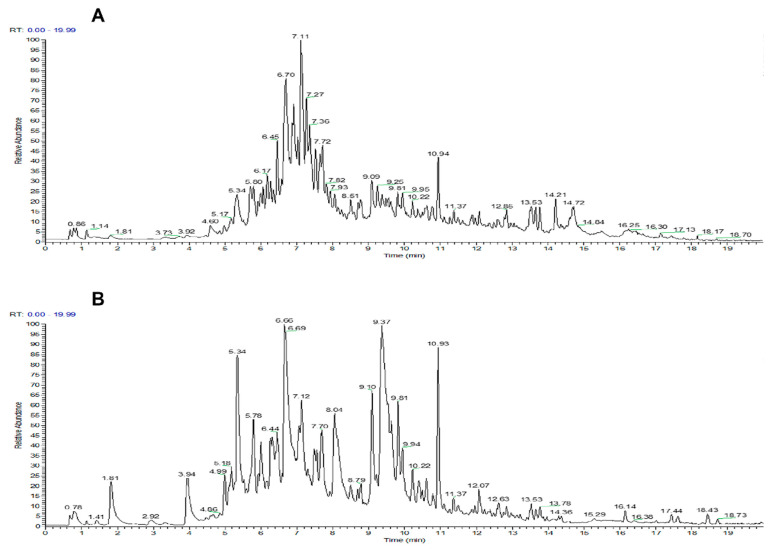
Total ion flow chromatograms of the purified flavonoids of *Polygonum perfoliatum* L. in (**A**) positive ion mode and (**B**) negative ion mode.

**Figure 10 molecules-30-00029-f010:**
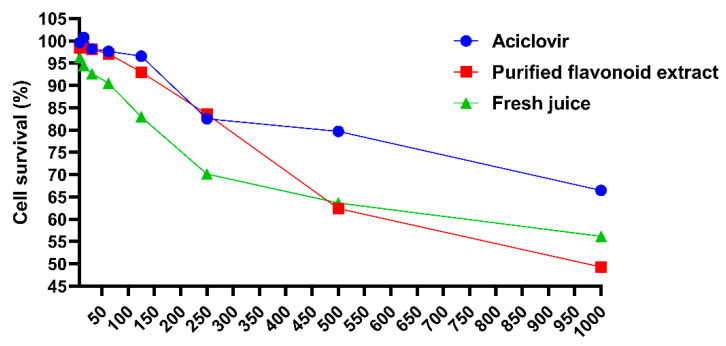
Effect of each dosing group on cell activity.

**Figure 11 molecules-30-00029-f011:**
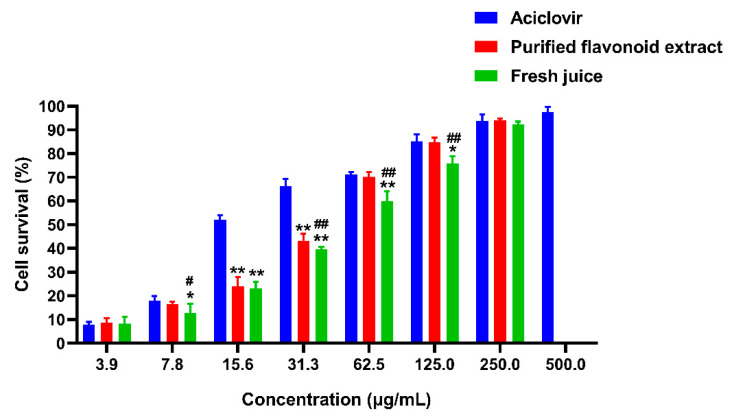
Anti-VZV activity of different dose groups of each drug (* *p* < 0.05, ** *p* < 0.01, compared with control; # *p* < 0.05, ## *p* < 0.01, compared with purified flavonoid).

**Figure 12 molecules-30-00029-f012:**
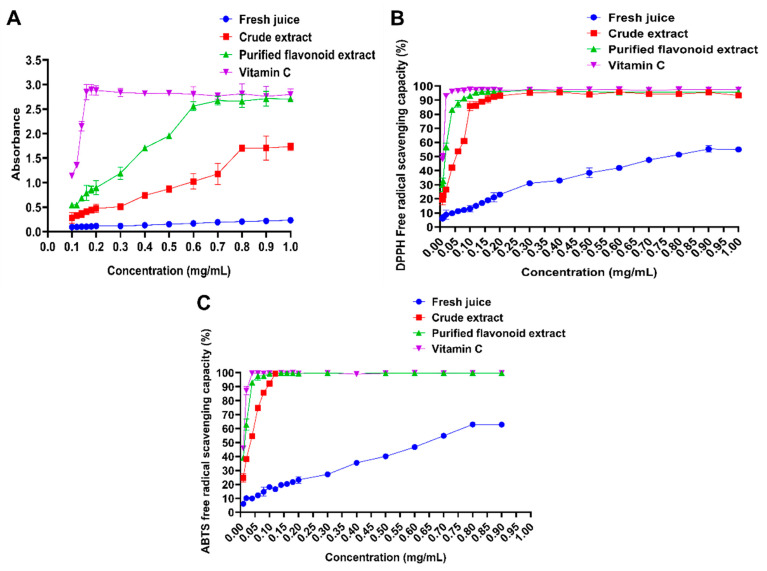
Antioxidant activity of flavonoids in *Polygonum perfoliatum* L. (**A**) Determination of total reducing capacity; (**B**) DPPH radical scavenging capacity; (**C**) ABTS radical scavenging capacity.

**Table 1 molecules-30-00029-t001:** Box–Behnken design and results.

Serial Number	*A*	*B*	*C*	Yield (mg/g)
1	−1	−1	0	13.4189
2	1	−1	0	13.0966
3	−1	1	0	12.2646
4	1	1	0	13.6247
5	−1	0	−1	12.8519
6	1	0	−1	12.3806
7	−1	0	1	13.0024
8	1	0	1	13.7025
9	0	−1	−1	12.9768
10	0	1	−1	11.6393
11	0	−1	1	13.0202
12	0	1	1	13.6123
13	0	0	0	14.6431
14	0	0	0	14.3891
15	0	0	0	14.5616
16	0	0	0	14.653
17	0	0	0	14.4096

**Table 2 molecules-30-00029-t002:** ANOVA of the response surface test regression model.

Source of Variance	Sum of Squares	Degree of Freedom	Mean Squared	F Value	*p*	Significance
Model	12.76	9	1.42	53.97	<0.0001	Significant
A	0.20	1	0.20	7.64	0.0280	
B	0.24	1	0.24	8.95	0.0202	
C	1.52	1	1.52	57.93	0.0001	
AB	0.71	1	0.71	26.95	0.0013	
AC	0.34	1	0.34	13.06	0.0086	
BC	0.93	1	0.93	35.44	0.0006	
A^2^	1.67	1	1.67	63.42	<0.0001	
B^2^	2.70	1	2.70	102.90	<0.0001	
C^2^	3.55	1	3.55	135.11	<0.0001	
Residual	0.18	7	0.026			
Lack of Fit	0.12	3	0.040	2.54	0.1946	Not significant
Pure Error	0.063	4	0.016			
Cor Total	12.94	16				
Std.Dev.	0.16			R-Squared	0.9858	
Mean	13.43			Adj R-Squared	0.9675	
C.V.%	1.21			Pred R-Squared	0.8433	
PRESS	2.03			Adeq Precision	22.601	

**Table 3 molecules-30-00029-t003:** Adsorption and desorption rates of flavonoids on different types of macroporous resins (*n* = 3, $\bar{x}$ ± s).

Macroporous Resin	Adsorption Rate (%)	Desorption Rate (%)
HPD-600	70.56 ± 1.53	45.81 ± 0.78
NKA-9	35.08 ± 0.97	72.31 ± 0.53
DM-130	27.56 ± 0.10	67.91 ± 0.91
AB-8	63.66 ± 0.87	86.45 ± 0.82
D-101	66.68 ± 0.67	92.87 ± 0.18
HPD-100	52.52 ± 2.37	83.63 ± 1.03
HP-20	56.27 ± 0.03	87.20 ± 0.74

**Table 4 molecules-30-00029-t004:** TCDI50 of VZV by CPE (*n* = 3, $\bar{x}$ ± s).

Virus Dilution	Lesion Hole	Non-Pathological Hole	Accumulate		Percentage of Lesion Holes Present (%)
			Lesion Hole	Non-Pathological Hole	
10^−1^	8 ± 0.00	0.00 ± 0.00	38.00 ± 1.00	0.00 ± 0.00	100 ± 0.00
10^−2^	8 ± 0.00	0.00 ± 0.00	30.00 ± 1.00	0.00 ± 0.00	100 ± 0.00
10^−3^	8 ± 0.00	0.00 ± 0.00	22.00 ± 1.00	0.00 ± 0.00	100 ± 0.00
10^−4^	8 ± 0.00	0.00 ± 0.00	16.00 ± 0.00	0.00 ± 0.00	100 ± 0.00
10^−5^	6.00 ± 1.00	2.33 ± 0.58	8.00 ± 0.00	2.33 ± 0.578	77.58 ± 4.20
10^−6^	2.00 ± 1.00	6.00 ± 1.00	2.00 ± 1.00	8.33 ± 0.578	19.09 ± 8.67
10^−7^	0.00 ± 0.00	8.00 ± 0.00	0.00 ± 0.00	16.33 ± 0.578	0.00 ± 0.00

**Table 5 molecules-30-00029-t005:** Response surface test factors and levels.

Variable	Level		
	−1	0	1
(A) Ethanol concentration	70	80	90
(B) Extraction temperature	80	90	100
(C) Solid-to-liquid ratio	20	30	40

**Table 6 molecules-30-00029-t006:** Physical parameters of different types of macroporous resins.

Macroporous Resin	Polarity	Surface Area (m^2^/g)	Pore Size (nm)
HPD-600	Polarities	550~600	80
NKA-9	Polarities	500~550	100~120
DM-130	Weak polarity	500~550	90~100
AB-8	Weak polarity	480~520	130~140
D-101	Non-polar	550~600	90~100
HPD-100	Non-polar	650~700	85~90
HP-20	Neutral polarity	550~600	90~100

**Table 7 molecules-30-00029-t007:** HPLC-MS/MS gradient elution procedure.

Time (min)	Flow Rate (mL/min)	B% (Acetonitrile)
0–2	0.3	5
2–6	0.3	30
6–7	0.3	30
7–12	0.3	78
12–14	0.3	78
14–17	0.3	95
17–20	0.3	95
20–21	0.3	5
21–25	0.3	5

## Data Availability

The data are included in the figures and tables in this manuscript.

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
