# Peer review of "Extraction and Purification of Flavonoids and Antiviral and Antioxidant Activities of Polygonum perfoliatum L."

_molecules, 2024, doi:10.3390/molecules30010029_

Round 1

Reviewer 1 Report

Comments and Suggestions for Authors

In this study, one-way analysis and response surface methodology were utilized to optimize the extraction process of total flavonoids from Polygonum perfoliatum L., refine the purification parameters, analyze the chemical composition of flavonoids, and evaluate their antiviral activity against varicella-zoster virus (VZV) as well as their antioxidant properties.  It can provide an optimized extraction and purification strategy for flavonoids from Polygonum perfoliatum L., along with evidence of their promising bioactivities. These findings offer technical support and a theoretical foundation for the further development and utilization of Polygonum perfoliatum L. resources. It is recommended to be published after minor revisions.

1. The visual representation of the data through figure is indistinct.

2. An absence of a contextual link can be observed in line 154. It stands out as something that doesn't have an obvious relation to the context in which it is placed.

3. The optimal conditions obtained in the model validation are inconsistent with the conclusion of the experimentally optimized best conditions based on Figure 1.

4. The letter cases in the figure captions are not consistent throughout the text.

5. For the experimental data of Anti-VZV activity, the specific source of the data is lacking.

6. According to the content of the article, section 2.6 is missing.

7. The format between numbers and units in the article is not consistent throughout.

8. The data in Table 6 is not mentioned throughout the text.

9. The format "TCID50" in line 665 of the article is inconsistent with the rest of the text.

10. The citation format of journal names in the references is not consistent throughout the text.

Reviewer 2 Report

Comments and Suggestions for Authors

Line 53–58: Why did you focus exclusively on comparing total yield when the primary compounds of interest are flavonoids? A higher or lower total yield does not necessarily correlate with higher or lower flavonoid content.

Line 67: It is unclear whether the RSM is applied to traditional extraction techniques or specifically to traditional Chinese herbal medicine. This distinction is important, as RSM can be used for optimizing traditional medicinal herb extractions and other methods, such as microwave-assisted extraction. Please revise for clarity.

Line 103: The type of extraction method being optimized is not specified. Consider revising this section, particularly the aim of the study, to better emphasize the significance and novelty of this research.

Results:

How were the flavonoid contents expressed? The manuscript only mentions "mg/g." Did you express them as equivalents of a specific flavonoid, such as catechin or rutin?

If possible, revise the term "liquid-to-material ratio" to "solid-to-liquid ratio" consistently throughout the text and graphs.

In Figure 1, there is an error where the term "liquid to solid" is used, followed by the unit "(g/mL)." This should be corrected.

Line 153: The sentence is incomplete. Please provide the missing portion for clarity.

Figures:

Figures 2, 5, 6, and 8 appear blurry. Please improve the image resolution for better readability.

Specifically, Figures  require immediate attention as their quality significantly affects the presentation of results.

Conclusion: The conclusion needs to be more comprehensive. Summarize the key findings of the study, highlight the significance of the research, and suggest potential applications or future directions for the work.

Overall Revision: The entire manuscript should be thoroughly reviewed for typographical errors, improved punctuation, and grammatical accuracy to ensure clarity.

Italic Style: Please review the entire manuscript to ensure consistent use of italic style where necessary, particularly for species names, variables, and technical terms.

Reviewer 3 Report

Comments and Suggestions for Authors

Major:

1. The authors should improve the quality of the presented figures; they are difficult to read and have low resolution, which makes it impossible to read the presented results.

2. The authors should standardize the notation of units; in the case of antioxidant activity studies, in one model, the authors use the notation ug/mL, and in the other, mg/mL; this should be standardized in the manuscript for consistency.

3. The presented abstract is too extended; I appreciate that the authors wanted to show all their results. However, the final version should be shortened as a summary.

4. The authors should expand the discussion of their results concerning the literature, which is currently limited.

Minor:

1. There is a persistent error in the manuscript text with a missing space before the citation

2. I would like to draw the author's attention to the fact that all phrases and abbreviations in Latin should be written in italics.

3. IC50 should be written with a lower index

Comments on the Quality of English Language

Minor linguistic and stylistic corrections are required.

Round 2

Reviewer 2 Report

Comments and Suggestions for Authors

The authors have revised the manuscript in accordance with the reviewers' comments.

Reviewer 3 Report

Comments and Suggestions for Authors

I thank the authors for their work in the revised manuscript and for fully addressing my earlier comments.

The revised manuscript is suitable for publication after making the minor corrections below.

Minor:

1. The end of the introduction section is written in the wrong style format

2. The Conclusion section is also written in the wrong style format